# The Multidimensional Logistic Model According to the Forecast of Employment of Graduates of Institutions of Higher Education of the Republic of Kazakhstan

Laura A. Khassenova [1], Altyn M. Yessirkepova [2],*[ID], Marat K. Seidakhmetov [3], Zarema A. Bigeldiyeva [4] and Dinmukhamed S. Zhakipbekov [5]

1   Department of Economics, M. Auezov South Kazakhstan University, Tauke Khan Av. 5,
    Shymkent 160001 , Kazakhstan
2   Cabinet Development of the Implementation of Educational Programs, Branch in Shymkent, Academy of
    Public Administration under the President of the Republic of Kazakhstan, Tauke Khan Av. 3,
    Shymkent 160001, Kazakhstan
3   Department of Management and Marketing, M. Auezov South Kazakhstan University, Tauke Khan Av. 5,
    Shymkent 160001, Kazakhstan
4   Department of Finance, M. Auezov South Kazakhstan University,
    Tauke Khan Av. 5, Shymkent 160001, Kazakhstan
5   Department of Accounting and Audit, M. Auezov South Kazakhstan University, Tauke Khan Av. 5,
    Shymkent 160001, Kazakhstan
*   Correspondence: essirkepova@mail.ru

**Abstract:** This article constructs a multidimensional logistic model for predicting the employment of graduates of higher educational institutions trained under the program of academic mobility, using the example of Kazakh universities. The purpose of the research is to identify the relationship between academic mobility and the employment of university graduates, as well as the necessary skills and competencies that can promote academic mobility in higher education institutions. This paper presents the results of calculating correlation coefficients and conducting a chi-square test, which demonstrated a relationship between the dependent variable and other questions in the graduate questionnaire. After the discovery of pairwise relationships, a multivariate logistic model was built that included statistically significant categories of responses. As a result, it was determined that the usefulness of participation in the academic mobility program is influenced by foreign language proficiency, an increase in academic mobility, the development of educational programs based on the needs of the labor market, as well as an increase in the weight of those methods that form practical skills in information analysis and the creation of public spaces in the labor market. The model demonstrated good predictive properties, which can be used to predict those individuals who were helped by academic mobility in finding employment.

**Keywords:** economics; academic mobility; employment of graduates; multidimensional model; correlation

**JEL Classification:** A2; C02; E01; I2; J6

## 1. Introduction

The purpose of this study is a comprehensive analysis of the role of academic mobility in increasing the competitiveness of university graduates in the labor market. To achieve this goal, the following research objectives were set:

- to concretize theoretical and methodological approaches to the implementation of academic mobility; to compare the approaches of foreign and domestic scientists in the research of problems;
- to explore academic mobility as an element of educational migration;

- consider the experience of foreign universities in implementing academic mobility programs and improving the competitiveness of graduates in the labor market;
- to reveal the main directions and features of the development of academic mobility in the Republic of Kazakhstan;
- based on the results of the survey of students, undergraduates and PhD doctoral students, identify positive trends and major shortcomings in the conduct of academic mobility;
- to analyze the implementation of the program of academic mobility for university graduates with the example of M. Auezov SKU, to reveal the role of academic mobility in increasing the competitiveness of university graduates in the labor market, and to identify measures that contribute to improving the competitiveness of university graduates in employment;
- to create a regression model corresponding to the type of data and questions that were included in the questionnaire, the purpose of which is to determine the factors influencing the benefits of participation in the academic mobility program and to identify the relationship between the factors and the benefits of internships in foreign universities.

The object of this study is the academic mobility and competitiveness of university graduates.

The subject of this study is the process of ensuring the competitiveness of graduates of Kazakhstani universities.

## 2. Literature Review

Trends in the development of higher and postgraduate education in Kazakhstan are focused on attracting foreign partners, students, scientists and academics. These trends are realized through academic and research collaboration, including student collaboration through participation in international and national projects, academic mobility of students and staff, and other factors that contribute to integration into the global educational space. International cooperation among Kazakh educational institutions is carried out in accordance with the legislation of the country and international agreements. International cooperation is represented by an appropriate legislative framework. This is the legal framework of the UN: the Universal Declaration of Human Rights (General Assembly UN 1948), the International Covenant on Economic, Social and Cultural Rights (General Assembly UN 1966a), the Convention against Discrimination in Education (General Assembly UN 1960), the ILO/UNESCO Recommendation Concerning the Status of Teachers (General Assembly UN 1966b), and the World Declaration on Higher Education for the 21st Century: Approaches and Practices (General Assembly UN 1998).

The internationalization of universities in the countries of the Bologna Process is regulated by documents of the Council of Europe (Bologna Process and Academic Mobility Center Ministry of Education and Science of the Republic of Kazakhstan 2019). This European Convention on the Equivalence of Diplomas Leading to Admission to Universities (ETS № 15) (Council of Europe 1953), European Convention on the Equivalence of Periods of University Study (Council of Europe 1956), European Convention on the Academic Recognition of University Qualifications (Council of Europe 1959), European Agreement on Continued Payment of Scholarships to Students Studying Abroad (Council of Europe 1969), Convention on the Recognition of Qualifications Concerning Higher Education in the European Region (Council of Europe 1997) and others.

The basis for interaction in the scientific and educational process is the following regulatory framework of the CIS—Agreement on Cooperation in Education (CIS 1999a), the agreement on cooperation in the training of scientific and scientific–pedagogical personnel and the nostrification of documents on their qualifications within the CIS (1992), the agreement on cooperation in the formation of a single (common) educational space of the CIS (1999a), and the Model Law on Education (adopted at the 13th Plenary Session of the Inter-Parliamentary Assembly of the CIS (1999b).

At the regional level, one of the most successful projects of internationalization in education has been the cooperation of member states of the Shanghai Cooperation Organization (SCO). The "SCO Universities" project aims to develop integration processes in education, science and technology, taking into account the best national traditions. The creation of the Central Asian Higher Education Area was initiated by the Taraz Declaration, adopted in 2007. One of the brightest examples of interaction is the «Central Asian Education Platform», implemented within the framework of the European Education Initiative for Central Asia. In June 2017, in Astana, the second meeting of the European Union and Central Asian education ministers was held. The main result of the meeting was the adoption of the Astana Declaration, which laid down a new stage in the process of strengthening ties between the countries of the Central Asian region and the European Union (Bologna Process and Academic Mobility Center of the Ministry of Education of the Republic of Kazakhstan 2018).

Kazakhstan, as a full member of the European Higher Education Area, has assumed the obligation to implement the parameters of the Bologna Process. Academic mobility is one of the important mandatory parameters of the Bologna Process.

In Kazakhstan, academic mobility is one of the most important forms of academic policy at universities, contributing to the internationalization of higher education.

In the Annex to the Recommendation of the Committee of Ministers of the Council of Europe (Strasbourg 1995), it is noted that "academic mobility refers to a period of study, teaching and/or research in a country other than the country of residence of the student or academic staff member. The training period is limited, after which the student or staff member must return to home country".

Børing et al. (2015) and Grossmann and Stadelmann (2013) investigated international academic mobility using a sociological method. The authors investigate a variety of mobility data from universities and research institutes. According to the results, 57% of university respondents and 65% of institutional respondents had experienced international mobility at least once in their research careers. International student mobility between universities strongly influences employment, which allows us to see the correlation between mobility and later being in demand in the labor market and is also an effective indicator that influences obtaining predictors of later mobility during a research and academic career.

The internationalization of the academic world is an important issue that has attracted public and political attention. The real information base on the migration and mobility of "academics", "scientists", "teachers", or "researchers" is still weak and insufficiently effective. Most of the available data in articles and publications focuses on international students and scientists, not on individuals moving for study or academic/research purposes. These data provide information only at a particular point in time, but there is no detailed information on the path of movement between countries. In recent years, the results obtained from surveys of "academic professions" and "researchers" have been conducted mainly in economically developed countries, mainly in Europe. These data provide insight into various forms of mobility, for example, pre-student migration, short-term student mobility and mobility for training programs, mobility for postgraduate students, professional mobility at various stages of professional careers, and shorter-term training programs related to academic and research work. All available information suggests that there are differences between countries and no features of convergence. Moreover, surveys confirm that international experience is often a valuable asset to academic research careers and employment.

Faruk (2016) noted that globalization affects the increase in international student mobility, leading to increased revenues for many top universities and host economies by providing educational and non-educational services to international students.

Davis and Goadrich (2006) note that logistic tools are a useful classical approach for solving problems related to regression analysis and its classification. ROC analysis is an essential tool for model quality analysis. This algorithm is actively used to build multidimensional models.

### 3. Methodology

The survey, processing and analysis of its results were carried out from May to August 2020. The survey was conducted using the technique of a personal, structured interview. A pilot study of 20 questionnaires was previously conducted and took place on 2 May 2020, among M. Auezov SKU university students.

Among the 14 universities in Kazakhstan that participated in the rating of the international rating agency Quacquarelli Symonds (QS), Auezov University is in the top three. It is the oldest and largest university in the Turkestan region. Due to the fact that this university has the largest number of students in a wide range of humanities and technical specialties, which positively affects the representativeness of such a university, and because of the convenience and available opportunities, the students of this university were identified by the basic sample. The survey control technique was also easier to provide on-site within the walls of the university.

The rest of the universities that were included in the sample were randomly selected from the list of universities in Kazakhstan, and a survey was organized at the locations of the selected universities.

The studied group consisted of students, undergraduates, doctoral students and graduates of 8 Kazakhstani universities who were trained under the academic mobility program at foreign universities. The sample population was 570 people.

The sample size was determined based on an estimate of the share of the trait in the general population using the following formula:

$$n = \frac{z^2 \cdot p(1-p)}{e^2} \tag{1}$$

where $z$ is the required confidence level (in this case 95%), $e$ is the permissible error of the sample study (in this case 0.03), and $p$ is the true estimated proportion of the trait in the general population (in this case 0.15—this proportion was determined from a working hypothesis; this is an estimate of the proportion of students who were trained under the academic program mobility). For students who were trained under the academic mobility program, it was based on the proportion of students at Mukhtar Auezov South Kazakhstan University.

$$n = \frac{1.96^2 \cdot 0.15(1-0.15)}{0.03^2} = 544 \tag{2}$$

Thus, the representative sample size should have been 544 people, but in the course of the study, 35 more people were interviewed. In total, 570 people were interviewed. These 8 universities were randomly selected from the entire population of universities in the country. Lists of persons who participated in the academic mobility program were obtained for these universities. Then the respondents were randomly selected using the random number generator function in Excel, were recruited and subsequently interviewed. Thus, the collected data can be considered representative of the entire population of universities in Kazakhstan.

To conduct the survey, a questionnaire was developed, which included 15 questions that described the respondents' activities in the field of academic mobility or their attitude to this activity.

All questions in the questionnaire relate to the nominal type of variables. The questionnaire lacked metric and rank scales, as well as variables characterizing the socio-demographic characteristics of respondents.

The answers to the questionnaire questions consisted of several categories. The respondent could choose one or more categories of answers. Since the questions related to the nominal scale, they were converted into binary variables that can only take two values of 1 or 0.

One of the questions concerned the benefits of participating in the academic mobility program for further employment of respondents. This question was identified as a target,

later this question acted as a dependent variable in the logistic regression, and all other questions were identified as influencing variables. Defining variables are presented in Table 1.

**Table 1.** Defining variables.

| Question Number in the Questionnaire | Variable Value | VARIABLE TYPE | Variable Assignment | Variable Encoding |
|---|---|---|---|---|
| q1 | Main advantages of AM | categorical | Independent | Binary 0 or 1 |
| q2 | Effective forms of AM | categorical | Independent | Binary 0 or 1 |
| q3 | Factors influencing the effectiveness of AM | categorical | Independent | Binary 0 or 1 |
| q4 | The quality of education | categorical | Independent | Binary 0 or 1 |
| q5 | Formidable AM barriers | categorical | Independent | Binary 0 or 1 |
| q6 | Benefits of participating in the academic mobility program | categorical | Dependent, target variable | Binary 0 or 1 |
| q7 | Impact on employment | categorical | Independent | Binary 0 or 1 |
| q8 | Measures to improve the efficiency of AM | categorical | Independent | Binary 0 or 1 |
| q9 | Measures to improve the competitiveness of graduates | categorical | Independent | Binary 0 or 1 |
| q10 | Measures necessary to prepare a competitive graduate for the labor market | categorical | Independent | Binary 0 or 1 |
| q11 | Measures to improve the effectiveness of the state policy on the employment of graduates | categorical | Independent | Binary 0 or 1 |
| q12 | Employment period | categorical | Independent | Binary 0 or 1 |
| q13 | Work in the specialty | categorical | Independent | Binary 0 or 1 |
| q14 | Causes of difficulties in the employment of graduates in the specialty | categorical | Independent | Binary 0 or 1 |
| q15 | The level of education at the university | categorical | Independent | Binary 0 or 1 |

*The Multidimensional Logistic Model According to the Forecast of Employment of Graduates Who Have Passed Academic Mobility*

A survey of graduates of universities in Kazakhstan was conducted in the course of this work concerning their attitude toward academic mobility. As a criterion for the effectiveness of academic mobility, the issue of its benefits for the employment of students was chosen. Most of the questions presented in the questionnaire fall into the category of questions with multiple responses, where a respondent could choose more than one response to a question.

The working hypothesis in this work is the existence of a relationship between the benefits of academic mobility and the questions in the questionnaire. Interpreting the categories of responses to these questions will help to understand what exactly influences the positive perception of academic mobility. This, in turn, will point to those aspects of the academic mobility system that are assessed by students as really important and on which development efforts should be accordingly focused.

To identify this relationship, correlation coefficients were calculated, and a statistical chi-square test was applied, which checks the presence of a relationship between qualitative variables. However, correlations and the chi-square test do not show the direction of this relationship; that is, they do not reveal a causal relationship. Logistic regression was applied to identify dependencies. These methods of analysis were chosen because the questions in the questionnaire refer to categorical data.

Analysis and data processing were carried out using the IBM SPSS Statistics (Statistical Package for Social Sciences), version 23 program—a statistical data processing computer program. This program has algorithms for the automatic selection of statistically significant variables, one of which was applied in this work (Teichler 2015).

## 4. Empirical Results and Discussion

First, we will carry out a correlation analysis, which will allow us to analyze the tightness of the relationship between the target variable "q6", which concerns the benefits of participating in the academic mobility program for further subsequent employment of the students, and the rest of the questions of the questionnaire. This question of the benefit of the respondent's participation in the academic mobility program was determined as the dependent variable in the logit model. In the case where the pair correlation is calculated between two dichotomous variables, it is possible to calculate the "phi" coefficient, Kramer's V, since their values coincide. Statistically significant correlation coefficients are marked with two asterisks (**), indicating 99% confidence, or one asterisk (*), indicating 95% confidence. Correlation coefficients can take on values from 1 to −1. Values close to zero show no relationship. If the correlation coefficient is positive, then the relationship between variables is directly proportional; that is, the dependent variable also increases with the increase of the independent variable. In addition, vice versa, if the correlation coefficient is negative, then the relationship between variables is inversely proportional, that is, the values of the dependent variable decrease as the values of the independent variable increase. The following table shows the correlation coefficients themselves, among which the largest values of positive correlation coefficients are identified. The greatest positive relationship was found between the dependent variable and such categories of responses as "Creation of offices under Government at the labor market", "employment within 1 year", "Reference level", "Academic mobility affected the development of abilities in research activity" and "Educational level and university ranking".

Correlation coefficients between the dependent variable and other questions in the questionnaire are presented in Table 2.

**Table 2.** Correlation coefficients between the dependent variable and other questions of the questionnaire.

| | q6 |
|---|---|
| Language immersion | 0.085 |
| Acquisition of fundamentally new knowledge and skills | 0.174 ** |
| Cultural exchange | −0.167 ** |
| Building international relations and contacts | 0.171 ** |
| Short stay at a foreign university for the purpose of exchanging experience and collecting information | −0.073 |
| Semester abroad | −0.180 ** |
| Summer semester (summer school) | −0.095 * |
| Joint research and development projects | 0.379 ** |
| Scientific conference (seminar course) | 0.372 ** |
| Educational level and university ranking | 0.411 ** |
| Degree of academic mobility organization and its duration | −0.273 ** |
| Ability to assimilate new competencies of university graduates | 0.171 ** |
| Reference level | 0.458 ** |
| The quality of education at my institution of higher education is higher than at a foreign one | −0.396 ** |
| The quality of education at my institution of higher education and a foreign institution of higher education is comparable | 0.290 ** |
| The quality of education at my institution of higher education is lower than at a foreign one | −0.142 ** |
| Language-specific | 0.107 * |
| Organizational (visa, completion of paperwork, domestic problems) | −0.128 ** |
| Resource-based (lack of financial and other resources) | −0.356 ** |
| Regulatory (difficulties in recognition of diplomas) | −0.065 |
| Content-related (comparability of content and level of programs) | 0.025 |
| Mobility helped overcome the language barrier | −0.316 ** |
| Gave the opportunity to deepen the knowledge gained and study new subjects | 0.130 ** |
| Competences acquired during the education helped (and will help) in further employment | −30.20 ** |
| Affected the development of abilities in research activity | 0.439 ** |
| Closer cooperation with potential employers and their interest in the employment of graduates who have passed academic mobility | −0.177 ** |
| Increasing the amount of funding from the state and from future employers | 0.009 |
| Increasing the amount of information about academic mobility in institutions of higher education | −0.209 ** |
| Extending the terms of academic mobility | 0.073 |
| Development of self-awareness in enrollees when choosing a profession or when choosing a specialty: focus on demand in the labor market | 0.079 |

**Table 2.** *Cont.*

| | q6 |
|---|---|
| Change of attitude toward reality from passively contemplative to actively transformational | −0.315 ** |
| Developing an educational program based on the needs of the labor market | 0.309 ** |
| Work with students in order to identify their needs for vocational education | −0.223 ** |
| Improving the activities of the institution of higher education as a buffer between graduates and employers, the implementation of advertising for its graduates | −0.151 ** |
| It is necessary to change the methods in the content of education, expanding the significance of those that form the practical skills of information analysis | 0.193 ** |
| The emerging role of independent work by students and the maximum increase in its volume | −0.213 ** |
| Strengthening relationships with practice, employers and research studies | −0.043 |
| Development of a system of quality indicators | 0.110 * |
| Strict quality control in education | −0.018 |
| Support for entrepreneurial initiatives by young people | −0.403 ** |
| Establishment of a state support service | −0.379 ** |
| Creation of offices under the Government in the labor market | 0.526 ** |
| Encouragement of employers creating workplaces for young people in the labor market | 0.301 ** |
| Within 3 months | −0.044 |
| Within 6 months | −0.158 ** |
| Within 1 year | 0.480 ** |
| More than 1 year | −0.266 ** |
| Yes | −0.168 ** |
| No | −0.148 ** |
| I go on to further study | 0.274 ** |
| No work experience | −0.231 ** |
| Lack of qualifications, lack of professional skills | 0.254 ** |
| No job vacancies | 0.254 ** |
| Not satisfied with salary | −0.050 |

Note: *—Statistically significant correlation coefficients indicating 95% confidence; **—Statistically significant correlation coefficients indicating 99% confidence, without *—Statistically significant correlation coefficient indicating confidence below 95%.

### 4.1. Chi-Square Test

The relationship between two variables related to the nominal or ordinal scale is tested using Pearson's chi-squared test, in which it is tested whether there is a significant difference between the observed and expected frequencies. When conducting a chi-square test, the mutual independence of two variables of the contingency table is tested, and, as a result, the dependence of both variables is indirectly revealed. The null hypothesis states that two variables are said to be mutually independent if the observed frequencies in the cells match the expected frequencies. If the observed and expected frequencies differ

statistically, then the null hypothesis is rejected and the alternative hypothesis is accepted, which states that the two variables are interdependent.

This test was conducted between the question about the benefits of academic mobility and other questions in the questionnaire. All tests showed a statistically significant result, which indicates the relationship between the question of the benefits of academic mobility and other questions of the questionnaire.

Pearson's chi-squared test results are presented in Table 3.

**Table 3.** Pearson's chi-squared test results.

|  |  | q6 |
|---|---|---|
| The main advantages of AM | Chi-square | 40.370 |
| | degrees of freedom | 5 |
| | Significance | 0.000 |
| Effective forms of AM | Chi-square | 151.204 |
| | degrees of freedom | 5 |
| | Significance | 0.000 |
| Factors influencing the effectiveness of AM | Chi-square | 240.149 |
| | degrees of freedom | 5 |
| | Significance | 0.000 |
| Quality of education | Chi-square | 159.773 |
| | degrees of freedom | 4 |
| | Significance | 0.000 |
| Insurmountable barriers of AM | Chi-square | 73.276 |
| | degrees of freedom | 5 |
| | Significance | 0.000 |
| Impact on employment | Chi-square | 270.751 |
| | degrees of freedom | 5 |
| | Significance | 0.000 |
| Measures to improve the efficiency of AM | Chi-square | 41.603 |
| | degrees of freedom | 5 |
| | Significance | 0.000 |
| Measures to increase the competitiveness of graduates | Chi-square | 129.021 |
| | degrees of freedom | 5 |
| | Significance | 0.000 |
| Measures necessary to prepare a competitive graduate for the labor market | Chi-square | 44.573 |
| | degrees of freedom | 5 |
| | Significance | 0.000 |
| Measures to improve the effectiveness of the state employment policy for graduates | Chi-square | 309.337 |
| | degrees of freedom | 5 |
| | Significance | 0.000 |
| Period of employment | Chi-square | 151.809 |
| | degrees of freedom | 5 |
| | Significance | 0.000 |

**Table 3.** Pearson's chi-squared test results.

|  |  | q6 |
|---|---|---:|
| Work in the specialty | Chi-square | 59.109 |
|  | degrees of freedom | 3 |
|  | Significance | 0.000 |
| Reasons for difficulties in finding graduates in the specialty | Chi-square | 89.409 |
|  | degrees of freedom | 5 |
|  | Significance | 0.000 |
| The level of education at the university | Chi-square | 41.329 |
|  | degrees of freedom | 3 |
|  | Significance | 0.002 |

From a simple statement of the fact of the existence of a relationship between variables, let us move on to forecasting and building a regression logistic model of dependency.

### 4.2. Building a Prognostic Model Based on Logistic Regression

The use of binary logistic regression will show whether it is possible to predict the values of the dependent variable depending on the values of one or more selected independent variables. The model has the form of an equation that contains the values of the regression coefficients.

The equation of the model is presented in the form of a logistic function, which has the following form:

$$p = \frac{e^y}{1 + e^y}. \tag{3}$$

Logistic regression calculates the probability of an event occurring; in our case, the probability that participation in an academic mobility program positively affected employment.

The decision was reached in six steps, each of which added a statistically significant variable to the model.

The most important result of modeling is the table with the calculated regression coefficients, which were selected by the program. For the selection of significant variables, the following selection method was used: direct step-by-step (conditional). As a result, this algorithm selected variables that have a statistically significant effect on the probability that participation in the academic mobility program helped respondents find employment. For the selected variables, regression coefficients were calculated, with the help of which it is possible to calculate the expected probability.

Regression coefficients with statistically significant variables are presented in Table 4.

**Table 4.** Regression coefficients with statistically significant variables.

| | | | B | Mean Square Error | Wald Test | Degree of Freedom | Significance | Exp (B) |
|---|---|---|---|---|---|---|---|---|
| **Step 6** | q3_4 | Foreign language proficiency | 2.031 | 0.379 | 28.768 | 1 | 0.000 | 7.620 |
| | q8_4 | Increasing the term of academic mobility | 4.484 | 0.643 | 48.598 | 1 | 0.000 | 88.625 |
| | q9_3 | Educational program development based on the needs of the labor market | 5.188 | 0.678 | 58.619 | 1 | 0.000 | 179.022 |
| | q10_1 | Methods in the content of training should be changed, expanding the weight of those that form practical skills for analyzing information | 4.177 | 0.647 | 41.689 | 1 | 0.000 | 65.200 |
| | q11_2 | Supporting students' entrepreneurial initiatives | −2.386 | 0.572 | 17.386 | 1 | 0.000 | 0.092 |
| | q11_4 | Creation of public places in the labor market | 3.804 | 0.514 | 54.883 | 1 | 0.000 | 44.899 |
| | Constant | | −6.914 | 0.843 | 67.221 | 1 | 0.000 | 0.001 |

Thus, the algorithm for selecting statistically significant variables stopped at a six-factor model. Five of six regression coefficients have a positive sign, suggesting a directly proportional relationship. Those respondents who noted that academic mobility was beneficial for their employment also noted the importance of the reference level; the extension of terms of academic mobility; developing an educational program based on the needs of the labor market; expanding the significance of those methods that form practical skills for analyzing information; and the creation of offices under the Government in the labor market. At the same time, although the support for entrepreneurial initiatives of young people has a statistically significant effect on the dependent variable, it has an adverse effect; that is, respondents who were helped by academic mobility in finding employment note that the support for entrepreneurial initiatives of young people is not important for employment. Perhaps the respondents do not perceive entrepreneurship as one of the types of employment.

The role of this assessment is to show by example how the probability of an affirmative answer to the benefits of participation in the academic mobility program is calculated, for what purpose logistic regression coefficients are calculated, and how they are used.

Let us make a forecast using the resulting model. The predicted value is determined by substituting the corresponding values of independent variables into the regression equation. The equation below shows the number of questions; their semantic meaning is given in Table

$$Y = -6.914 + 2.031 * q3\_4 + 4.484 * q8\_4 + 5.188 * q9\_3 + 4.177 * q10\_1 - 2.386 * q11\_2 + 3.804 * q11\_4 \quad (4)$$

If we take the values of six independent variables for the resulting model for the first respondent from our database and substitute them into the resulting model, then we obtain:

$$Y = -6.914 + 2.031 \cdot 0 + 4.484 \cdot 0 + 5.188 \cdot 0 + 4.177 \cdot 1 - 2.386 \cdot 1 + 3.804 \quad (5)$$

The Y value will be −5.123 given the values of the independent variables.
This value must be substituted into the logistic function.
Let us substitute the resulting value into the formula:

$$p = \frac{e^{-5.123}}{1 + e^{-5.123}} \quad (6)$$

The result is 0.00593. This value indicates the probability that participation in the academic mobility program helped in finding a job. In this case, the forecasting results coincided with the actual value indicated by the respondent, namely, that the academic mobility program helped the respondent find a job.

The quality of the built model and its predictive power can be judged by several of the following indicators:

- Pseudo R-square;
- Percentage of cases correctly predicted;
- ROC curve.

R-square for model are presented in Table 5.

**Table 5.** R-square for the model.

| Step | $-2$ Log Likelihood | Cox–Snell R-Square | Nagelkirk R-Square |
|:---:|:---:|:---:|:---:|
| 6 | 216.208a | 0.581 | 0.787 |

Measures of certainty are one of the indicators of the quality of the built model: Cox–Snell R-square and Nagelkirk R-square, which are R-squared counterparts of linear regression models. They also indicate the percentage of dependent variable variation that the model can clarify. At the same time, the indicator of Nagelkirk R-square is more perfect, which indicates that the model can clarify the behavior of the dependent variable by 78.7%.

The following classification table can also be called a measure of the quality of the built model. It shows the number and percentage of correctly predicted dependent variable values. The percentage of correctly predicted values was 90% for the resulting model.

Model classification table are presented in Table 6.

**Table 6.** Model classification table.

| | | | Predicted | | |
|:---:|:---:|:---:|:---:|:---:|:---:|
| | **Observed** | | **q6** | | **Percentage of Correct** |
| | | | **0** | **1** | |
| Step 6 | q6 | 0 | 248 | 30 | 89.2 |
| | | 1 | 15 | 166 | 91.7 |
| | Total percentage | | | | 90.2 |

There is another indicator that can be used to evaluate the discriminatory power of logistic regression. The ROC curve (Eng. receiver operating characteristic) allows us to evaluate the efficiency of a binary classifier and select the optimal cutoff threshold. It represents a curve of the ratios of correct and false detections of students who noted the usefulness of academic mobility, obtained by varying the cutoff point of the probability of occurrence. It will be recalled that the cutoff value is 0.5 by default. The same value was set by default when building the logistic regression. The cutoff point does not affect the values of the regression coefficients.

The ROC curve plot is constructed as follows: False positive cases (FPCc) are plotted along the x-axis of the ROC curve (1 − specificity). True positive cases (TPCs) are plotted along the y-axis of the ROC curve (sensitivity). The ROC curve plot passes through the upper left corner in the case of perfect classification. In this case, the proportion of true positive cases is 100%, and the proportion of false positive cases will be 0%. Therefore, the closer the curve is to the upper left corner, the higher the discriminative power of the model. In addition, vice versa, the smaller the bend in the curve and the closer it is to the diagonal line (corresponding to the useless classifier), the model is less efficient.

An optimal model should have 100% sensitivity and 100% specificity, but this is impossible to achieve. In practice, an ROC curve is constructed—a curve of the ratio of

true positive cases (sensitivity) and false positive cases (1 − specificity) for various cut-off thresholds—and such a cut-off threshold is chosen that gives optimal sensitivity and specificity values.

ROC curve is presented in Figure 1.

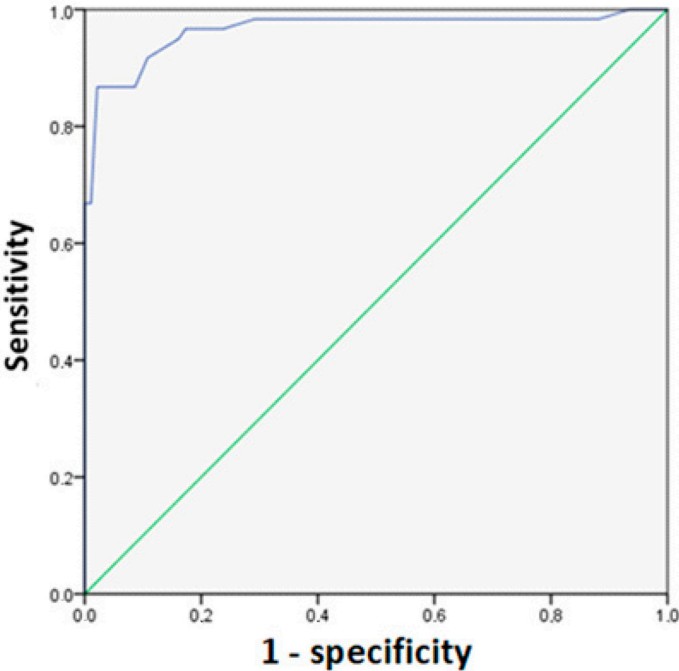

**Figure 1.** ROC curve.

The plots show that the curve deviates from the diagonal line, which indicates good discriminative power of models. An indicator of this power is the area under the ROC curve, which is 0.5 for a model with zero predictive power and 1 for a model with the highest predictive power. The model resulted in a value equal to 0.965, which allows rejecting the null hypothesis that the area under the ROC curve does not differ from 0.5 and that the model has low predictive power and accepting an alternative hypothesis, according to which the model well clarifies the variation of the dependent variable and has good predictive power.

Area under curve is presented in Table 7.

**Table 7.** Area under curve.

| Region | Standard Error | Asymptotic Value.b | Asymptotic 95% Confidence Interval | |
| --- | --- | --- | --- | --- |
| | | | Lower Bound | Upper Bound |
| 0.965 | 0.010 | 0.000 | 0.946 | 0.984 |

The following table lists the values for "Sensitivity" and "1 − Specificity". The default cut-off points of 0.5 was chosen in our logistic regression model. The value of sensitivity or TPCs with it is 0.89; that is, 89% of respondents who indicated that academic mobility helped them in employment were correctly predicted by the model; however, 8.6% of respondents who indicated that academic mobility did not help them in employment were mistakenly identified as persons whom it, on the contrary, helped in employment.

Area under ROC Curve are presented in Table 8.

**Table 8.** Area under the ROC Curve.

| True If Greater Than or Equal To | Sensitivity | 1 − Specificity |
|:---:|:---:|:---:|
| 0.0000000 | 1.000 | 1.000 |
| 0.0003940 | 1.000 | 0.957 |
| 0.0008448 | 1.000 | 0.935 |
| 0.0025416 | 0.983 | 0.881 |
| 0.0050094 | 0.983 | 0.871 |
| 0.0067235 | 0.983 | 0.561 |
| 0.0077799 | 0.983 | 0.529 |
| 0.0120765 | 0.983 | 0.399 |
| 0.0294187 | 0.983 | 0.367 |
| 0.0430989 | 0.983 | 0.324 |
| 0.0508268 | 0.983 | 0.313 |
| 0.0595252 | 0.983 | 0.291 |
| 0.0709177 | 0.967 | 0.237 |
| 0.0959550 | 0.967 | 0.216 |
| 0.1310098 | 0.967 | 0.194 |
| 0.1811458 | 0.967 | 0.183 |
| 0.2709151 | 0.967 | 0.173 |
| 0.3382006 | 0.950 | 0.162 |
| 0.4310759 | 0.917 | 0.108 |
| 0.5459719 | 0.867 | 0.086 |
| 0.6233448 | 0.867 | 0.054 |
| 0.7076803 | 0.867 | 0.043 |
| 0.7726774 | 0.867 | 0.022 |
| 0.8264140 | 0.669 | 0.011 |
| 0.8702456 | 0.669 | 00.000 |
| 0.9145725 | 0.420 | 00.000 |
| 0.9486140 | 0.403 | 00.000 |
| 0.9777334 | 0.066 | 00.000 |
| 0.9992437 | 0.050 | 00.000 |
| 1.0000000 | 0.000 | 0.000 |

Sensitivity is the number of true positive cases divided by the total number of positive cases in the sample. Sensitivity is also called completeness. It is measured according to the formula:

$$\text{Sensitivity} = \frac{\text{TPCs}}{\text{TPCs} + \text{FPCs}} \qquad (7)$$

where TPCs are true positive cases and FPCs are false positive cases.

$$\text{Sensitivity} = \frac{166}{166 + 15} = 0.917 \qquad (8)$$

In our example, sensitivity is the power of the model to correctly detect cases where respondents positively assessed the impact of academic mobility on student employment. A model with high sensitivity maximizes the proportion of correctly classified cases.

$1 -$ specificity (one minus specificity) is the number of false positive cases (FPCs) divided by the total number of true negative cases (TNCs) in the sample and is calculated according to the formula:

$$1 - \text{specificity} = \frac{\text{FPCs}}{\text{FPCs} + \text{TNCs}} \tag{9}$$

or

$$1 - \text{specificity} = \frac{30}{30 + 248} = 0.108 \tag{10}$$

In our example, $1 -$ specificity characterizes the level of "false responses" of the model when persons who have been helped by academic mobility in employment are classified as persons whom it has not helped in employment.

Thus, calculating the correlation coefficients and performing the chi-square test given above showed the existence of a relationship between the dependent variable and other questions of the questionnaire. After the detection of pairwise relationships, a multidimensional logistic model was built that included statistically significant categories of responses. The model showed good predictive properties. Accordingly, this model can be used to predict the probability that participation in the academic mobility program helped in finding a job.

## 5. Concluding Remarks

The analysis of correlation coefficients and the chi-square test showed a relationship between the dependent variable and some categories of questions in the questionnaire. However, not all categories of questionnaire responses that showed a pairwise relationship with the dependent variable were included in the logistic regression model. For example, increasing the terms of academic mobility, which showed no significant pairwise correlation coefficient with the question about the benefits of academic mobility, was useful in predicting the dependent variable and entered the logistic regression model.

After finding pairwise relationships, a multidimensional logistic model was constructed that included such statistically significant categorical independent variables as the following: foreign language proficiency; increasing the term of academic mobility; educational program development based on the needs of the labor market; increasing the weight of those methods that form practical skills of information analysis; and the creation of public places in the labor market. According to the value of the regression coefficient of the model, the greatest influence on the benefits of participation in the academic mobility program is the development of an educational program based on the needs of the labor market. In addition to this aspect, a stronger impact on employment benefits comes from increasing the length of the program, developing practical information analysis skills, and creating jobs at the expense of the state. Thus, these aspects of the development of the academic mobility program are of the utmost importance from the point of view of the study participants. In other words, those respondents who chose these response categories during the survey were also more likely to indicate that participation in the academic mobility program helped them in their employment, i.e., these aspects of education should, in their view, have a positive impact on the employment of students who took part in the academic mobility program. At the same time, we can say that for those students for whom these categories of answers did not seem important, participation in the academic mobility program did not help in their future employment. The model demonstrated good prognostic properties. Accordingly, it can be concluded that academic mobility had an impact on their employment. The presented model shows the close connection between the passages of academic mobility and how much universities will organize them qualitatively for their students, the more successfully they will advance in the labor market and contribute to the development of the country and economic sectors. There are questions related to the quality of higher education and its impact on the labor market and economic development, but we think that this will be the topic of our next study.

**Author Contributions:** Conceptualization, L.A.K. and A.M.Y.; technique, L.A.K. and A.M.Y.; software, M.K.S.; validation, Z.A.B. and D.S.Z.; formal analysis, L.A.K. and A.M.Y.; investigation, L.A.K., A.M.Y., M.K.S., Z.A.B. and D.S.Z.; resources, M.K.S.; data supervision, L.A.K.; writing—rough preparation, L.A.K. and A.M.Y.; writing—review and editing, Z.A.B. and D.S.Z.; visualization, M.K.S.; supervision, L.A.K. and A.M.Y.; project administration, A.M.Y. All authors have read and agreed to the published version of the manuscript.

**Funding:** This research received no external funding.

**Informed Consent Statement:** Not applicable.

**Data Availability Statement:** Not applicable.

**Conflicts of Interest:** The authors declare no conflict of interest.

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
