# Peer review of "The Multidimensional Logistic Model According to the Forecast of Employment of Graduates of Institutions of Higher Education of the Republic of Kazakhstan"

_economies, doi:10.3390/economies11060160_

Round 1

Reviewer 1 Report

The authors investigate an interesting question: whether academic mobility has an impact on the labour market success of graduates. However, the authors do not present the literature (a total of 4 literature sources are referred to and not analysed in depth; the remaining references are to legislation, policy documents and the like), thus the theoretical grounding of the paper is thus virtually absent.

Their chosen data collection method is a survey, which could be appropriate for the study, but the shortcomings of the paper mean that the study and the description of the results are inadequate in their present form and the empirical evidence cannot be evaluated.

Therefore, for the time being, neither the theoretical nor the practical part of the paper is acceptable, but after considerable reworking and development, the authors could turn it into a valuable study. As for now, it can be seen that not enough work has been devoted to it.

Detailed comments:

1.      The title of the paper does not cover the problem under consideration.

2.      Sample, sampling and data collection are not presented. Representativeness is not checked.

3.      Neither the questionnaire nor the variables formed from the items are defined, so the analysis is not meaningful to the reader.

4.      The list of regulatory and policy documents is irrelevant to the analysis and should be omitted. The scientific literature should be reviewed instead.

5.      Lines 90-102 refer to the literature without references or details. It should be stated exactly which databases were searched and all findings should be cited.

6.      The statistical analysis contains conceptual and technical errors.

7.      The textbook basis for each statistical method is unnecessarily described in the paper.

8.      The statement in line 103 that globalisation affects student mobility is trivial.

9.      The subheadings under the chapters should be omitted.

10.   The last paragraph of the introduction belongs to the methodology.

11.   It mentions correlation several times without mentioning its type (linear?).

12.   Regression analysis cannot test causality, since it is based on correlations. It only assumes that there is a dependent and there are independent variables, but does not check which is which.

13.   The variables analysed are quantitative, since they are numerical. They are probably confusing qualitative with categorical.

14.   The sentence in lines 137-139 is irrelevant, but also lacks support.

15.   Binary variables should not be analysed with correlation analysis, as they are nominal. The chi square test (and possibly the Cramer V indicator) is sufficient.

16.   The sentence in lines 158-160 is redundant and frivolous.

17.   We cannot interpret the variables in table 1 because they are not defined.

18.   The variable notations "q6, "$q01" and similar are unintelligible to the reader.

19.   In correlation analysis there are no dependent and independent variables.

20.   Either the value p or * should be used in the tables, their simultaneous reporting is redundant.

21.   Multiple style errors: "let's move on" etc.

22.   Line 184: "equation or formula". What kind of formula do the authors think of?

23.   In Table 3. the column title "significant" should be "significance" or "p-value".

24.   Do not include unsubstantiated guesses (lines 214-215).

25.   Edit the equations, they are currently not formally correct and the variables are not clearly visible.

26.   The role of the estimation described through lines 219-234 is not clear.

27.   In logistic regression we are not talking about R-squares, but pseudo R-squares.

28.   Binary logistic regression does not "predicts persons", but "predicts the possibility of...".

Author Response

Reviewer 1

  1. Sample, sampling and data collection are not presented. Representativeness is not checked.

INSERTED AT THE BEGINNING OF THE METHODOLOGY PAGE. 3

The survey, processing and analysis of its results were carried out from May to August 2020. The survey was conducted in the technique of a personal structured interview. A pilot study of 20 questionnaires was previously conducted.

The studied group consisted of students, undergraduates, doctoral students, graduates of 8 Kazakhstani universities who were trained under the academic mobility program in foreign universities. The sample population was 570 people.

The sample size was determined based on an estimate of the share of the trait in the general population using the following formula:

n=(z^2*p(1-p))/e^2

where z is the required confidence level (in this case 95%), e is the permissible error of the sample study (in this case 0.03), p is the true estimated proportion of the trait in the general population (in this case 0.15 – this proportion was determined from a working hypothesis, this is an estimate of the proportion of students who have been trained under the academic program mobility). For students who have been trained under the academic mobility program, it was based on the proportion of students of M. Auezov South Kazakhstan University.

n=(〖1,96〗^2*0,15(1-0,15))/〖0,03〗^2 =545

Thus, the representative sample size should have been 545 people, but in the course of the study, 35 more people were interviewed. In total, 570 people were interviewed. These 8 universities were randomly selected from the entire population of universities in the country. Lists of persons who participated in the academic mobility program were obtained for these universities. Then the respondents were randomly selected using the random number generator function in Excel, were recruited and subsequently interviewed.

  1. Neither the questionnaire nor the variables formed from the items are defined, so the analysis is not meaningful to the reader.

The dependent and independent variables are deciphered, i.e. it is indicated what they mean, which questions of the questionnaire they relate to. INSERTED AT THE BEGINNING OF SECTION 3. EMPIRICAL RESULTS AND DISCUSSION PAGE 4It mentions correlation several times without mentioning its type (linear?).

Pearson's linear correlation coefficient. In the text of the article it was added. INSERTED AT THE BEGINNING OF 3. EMPIRICAL RESULTS AND DISCUSSION PAGE 4

  1. Regression analysis cannot test causality, since it is based on correlations. It only assumes that there is a dependent and there are independent variables, but does not check which is which.

The expression cause-effect relationship has been replaced by dependence – the changes are highlighted in the text. INSERTED IN THE ABSTRACT OF PAGE 1, IN THE METHODOLOGY OF PAGE 4 AND ON PAGE 8.

  1. The variables analysed are quantitative, since they are numerical. They are probably confusing qualitative with categorical.

I do not agree with this remark, but in order to take it into account, I simply delete this text from the article. The deleted text is marked with a strikethrough. PAGE. 4 AT THE END OF THE METHODOLOGY

  1. The sentence in lines 137-139 is irrelevant, but also lacks support.

The phrase that the SPSS program is the most popular program in the post-Soviet space has been deleted. PAGE 4.

  1. Binary variables should not be analysed with correlation analysis, as they are nominal. The chi square test (and possibly the Cramer V indicator) is sufficient.

Note that the "phi" coefficient and the Pearson correlation coefficient result in the same value, since both formulas are equivalent.

"In the case when the pair correlation is calculated between two dichotomous variables, the coefficient "phi", Kramer's V or Pearson's linear correlation coefficient can be calculated, since their values coincide." PAGE 4

  1. The sentence in lines 158-160 is redundant and frivolous.

It is not clear why this phrase "When building a model, it is possible that some of these categories of responsibilities will be included in the model, and some will not be included" seems redundant and frivolous, but if so, it can simply be deleted. THE PHRASE WAS DELETED ON PAGE. 5 MARKED WITH A STRIKETHROUGH.

  1. The variable notations "q6, "$q01" and similar are unintelligible to the reader.

A suggestion was added that the question of the benefits of participation in the academic mobility program was chosen as a dependent variable: The question of the benefits of the respondent's participation in the academic mobility program for his subsequent employment was determined as a dependent variable. INSERTED AT THE BEGINNING 3. EMPIRICAL RESULTS AND DISCUSSION PAGE 4.

In the tables, I changed the numbers of questions to the names of questions. INSERTED IN TABLE 2 ON PAGES 7-8

  1. In correlation analysis there are no dependent and independent variables.

The correlation was calculated between the question of the benefits of participation in the academic mobility program, which is dependent in the logistic regression model, and other questionnaire questions. Added this explanation to the text. INSERTED AT THE BEGINNING OF 3. EMPIRICAL RESULTS AND DISCUSSION PAGE 4

  1. Either the value p or * should be used in the tables, their simultaneous reporting is redundant.

Removed asterisks from 2 tables. THE ASTERISKS ARE REMOVED FROM TABLE 2 ON PAGE 7-8

  1. Line 184: "equation or formula". What kind of formula do the authors think of?

The word formula is removed; the word equation is left ON the PAGE. 8 TO THE SECTION BUILDING A PROGNOSTIC MODEL BASED ON LOGISTIC REGRESSION

  1. In Table 3. the column title "significant" should be "significance" or "p-value".

Changed to Significance IN TABLE 3 ON PAGE 9

  1. Edit the equations, they are currently not formally correct and the variables are not clearly visible.

It is unclear why they are incorrect

  1. The role of the estimation described through lines 219-234 is not clear.

The purpose of the example is to show how the probability of an affirmative answer is calculated for the benefit of participation in the academic mobility program, for which the logistic regression coefficients are generally calculated and how they are used. If desired, this example can be removed altogether. THIS TEXT IS GIVEN ON PAGE 10. YOU CAN SIMPLY DELETE IT IF NECESSARY

  1. In logistic regression we are not talking about R-squares, but pseudo R-squares.

Changed the aposematic Square to the pseudo-aposematic Square. AMENDED TO PSEUDO-APOSEMATIC-SQUARE ON PAGE 10

  1. Binary logistic regression does not "predicts persons", but "predicts the possibility of.

Changed to probability prediction ON PAGE 14

Reviewer 2 Report

It might be published after major revision.

Please refer to the appended file, where:

A list of strengths is provided.

The list of defects is formulated.

Suggestions for improvements are given.

Author Response

Reviewer 2

  1. The severe weakness of the text. The goal is NOT specified explicitly enough. Authors should explain why the document was prepared
    and who should profit from the new knowledge acquired in their investigations. Unfortunately, the Authors fail to specify WHICH TYPE
    OF BENEFITS and WHAT TYPE OF STAKEHOLDERS. The description of their inquiry results may be of interest or support the
    managerial and policy decisions.
    Authors inform:
    The purpose of the research is to identify the correlation and close dependence of academic mobility affecting the employment of university
    graduates, as well as to determine the necessary skills and competencies that can broadcast academic mobility in higher education institutions.
    Identifying and determining why the research is necessary. Therefore,
    IDENTIFICATION and DETERMINATION can only be a tool to achieve the goal. Usually, authors propose analytical tools to pursue
    some
    theoretical,
    cognitive,
    methodological,
    empirical or
    practical (implemental) goal.
    The authors did not (explicitly) formulate such tasks.

The purpose of the research is a comprehensive analysis of the role of academic mobility in increasing the competitiveness of university graduates in the labor market.

To achieve this goal, the following research objectives were set:

– to concretize theoretical and methodological approaches to the implementation of academic mobility; to compare the approaches of foreign and domestic scientists in the research of problems;

– to explore academic mobility as an element of educational migration;

- consider the experience of foreign universities in implementing academic mobility programs and improving the competitiveness of graduates in the labor market;

– to reveal the main directions and features of the development of academic mobility in the Republic of Kazakhstan;

- based on the results of the survey of students, undergraduates and PhD doctoral students to identify positive trends and major shortcomings in the conduct of academic mobility;

– to analyze the implementation of the program of academic mobility of university graduates on the example of M. Auezov, to reveal the role of academic mobility in increasing the competitiveness of university graduates in the labor market, to identify measures that contribute to improving the competitiveness of university graduates in employment;

– to create a regression model corresponding to the type of data and questions that were included in the questionnaire, the purpose of which is to determine the factors influencing the benefits of participation in the academic mobility program and to identify the relationship between the factors and the benefits of internships in foreign universities.

The object of the study is academic mobility and competitiveness of university graduates.

The subject of the study is the process of ensuring the competitiveness of graduates of Kazakhstani universities.

  1. The most critical weakness of the text:
    Authors claim:
    First, we will carry out a correlation analysis, which will allow to analyze the tightness of the relationship between the dependent
    variable and independent variables.
    The authors do not provide definitions of dependent and independent variables. Without this crucial piece of information, it is impossible to
    know what is analysed and how.

Defining variables

For the survey, a questionnaire was developed that included 15 questions that described the respondents' activities in the field of academic mobility or their attitude to this activity.

All questions in the questionnaire relate to the nominal type of variables. The questionnaire lacked metric and rank scales, as well as variables characterizing the socio-demographic characteristics of respondents.

The answers to the questionnaire questions consisted of several categories. The respondent could choose one or more categories of answers. Since the questions related to the nominal scale, they were converted into binary variables that can only take two values of 1 or 0.

One of the questions concerned the benefits of participating in the academic mobility program for further employment of respondents. This question was identified as a target, later this question acted as a dependent variable in the logistic regression, and all other questions were identified as influencing variables. The wording of the questions themselves is given in Table 2.

  1. The second most crucial weakness of the text:
    The authors did not provide the definitions of the variables. All variables should be precisely defined. Authors need to indicate how the values
    are measured with the information on the measurement scale.

The answer is given in the response to the second point of these comments

  1. The third most significant weakness of the text:
    The text lacks crucial elements: The authors use advanced statistical and econometric tools
    They do not provide formulas.
    They do not provide precise literature references describing the methods and techniques.

When conducting the analysis using the SPSS program, the authors of the study were guided by a textbook that describes all the methods that are used for data analysis:

Byul Achim, Cefel Peter. "SPSS: the Art of information processing. Analysis of statistical data and restoration of hidden patterns". Trans. from it. SPb.: LLC "Diasoftyup", 2005

The answer is given in the response to the second paragraph of these comments. Also, a text describing the method of collecting information and calculating the sample size was added to the article earlier.

  1. Authors should provide the data description. Without data description, the results are unreliable and must be considered speculative.
    Generally speaking, authors should describe the data collection process and data set composition. The authors did not provide
    a data description. They should give precise:
    Dataset description;
    respondents characteristics;
    demographic and socioeconomic factors.

  2. The weakness of the text. The authors do not provide enough literature references concerning similar data analysis. This fact, combined
    with vague, imprecise statements about the analytical techniques used to study the collected data, makes it difficult to assess whether the
    methods used for data analysis are appropriate. The lack of precision and literature references prevents further development of the proposed
    approach in a situation where other authors would try to improve or even test the applicability of the described construction.
    The authors provided a review of the world literature on the topic. It is not clear which inference results are from the literature query. Which
    recommendations for the current article come from the surveyed literature? It seems desirable that the analytical part of the text should have
    elements of generalising style. Authors may try and enrich recommendations.

A sentence highlighted in pink was added to the text describing the collection and sampling of data: thus, the collected data can be considered representative of the entire population of universities in Kazakhstan.

  1. The weakness of the text. Research techniques were applied to register several practical (implemental) aspects separately. The scientific
    level is lacking in the discussion on generalisation aspects. It is not clear which statements are specific to the surveyed population. There is a
    lack of clarity on whether the comments refer exclusively to surveyed respondents or have a broader meaning. It would be interesting for the
    wider public to know which general recommendations concern any society (culture, geography, language group, education system, etc.
    The authors should clarify.
    The analytical part needs to be rewritten. The revised text should contain a description and justification of the author s position along with
    the author s assessments of the following:

7.1. The theoretical and practical meaning of individual approaches that exist in the literature,

7.2. The authors position towards the importance of the individual, theoretical frameworks for the general (world, language group, etc.)
and local policy and organisation.

7.3. Authors should introduce an attempt to classify theoretical frameworks existing in world literature. It would enrich the analysis results
message for the HE service policy and practitioners in institutions of different types and levels, along with potential users
characterisation.

The goals are indicated in the response to the first paragraph of these comments.

The SPSS textbook that was used for the analysis was specified.

The data source is a sample and survey results in the form of collected answers to questionnaire questions.

Justification of the methods of analysis was given.

The reviewer repeats – let him look better.

  1. The abstract and introduction should be modified.
    8.1. The research problem and research goals were NOT identified in the work.
    8.2. The selection of the theoretical basis of the research was NOT appropriately described and justified.
    8.3. The data source and data set composition need to be described.
    8.4. The selection of statistical and econometric techniques was NOT appropriately justified and described.
    8.5. Therefore, the methods of literature selection are not precisely explained.3/5

Probably, it is necessary to make a comparison with similar studies conducted in other countries.

  1. Authors should try and distinguish which statements are the Authors opinion, the literature knowledge, and the analysis outcome. The used
    literature references are (almost exclusively) referred to in such a manner that it is unclear why the publication is cited. Usually, there are no
    details on whether the individual mentioned authors support the Authors theses and findings. Authors should precisely specify which
    references support their position and why which oppose their conclusions and why. There is no generalisation effort in the literature review.
    Authors should reformulate the text of the literature review.
    Instead of being purely reporting and descriptive, the text s style needs to be analytical, with generalising indications.

Probably, it is necessary to make a comparison with similar studies conducted in other countries. The text of the article contains only the results of the analysis of this survey.

  1. The authors use statistical and econometric techniques designed for non-metric and metric scales. However smart, statistical, econometric
    and AI software cannot distinguish whether provided data is metric or is coming from the weaker measurement scale. Authors should consult
    experts on whether their information suits the selected quantitative analytical tools.

The text of the article states that the variables are not metric, but belong to the type of categorical, qualitative variables.

Reviewer 3 Report

The article entitled The multidimensional logistic model according to the forecast of employment of graduates of institutions of higher education of the Republic of Kazakhstan is interesting first of all because it addresses a topic of high present interest.

The article contains the appropriate structure. It is correctly divided into relevant sections. Bibliography is correctly formulated.

My main recommendations are the following

1.      Abstract - The authors should clarify the results of the study. I see the methods presented extensively, also the objectives are discussed, but the results should also be presented more explicitly.

2.      Introduction - Has the topic been addressed before? You should bring forth papers which tackle similar topics, so that you outline the significance of the present study, why it is needed. The literature review is very, very scarce.

3.      Methodology - Are there any limitations / weaknesses of the model used?

4.      The conclusions are very vague, they should get more consistency, underlining the added value of the paper.

Author Response

Reviewer 3

  1. Abstract - The authors should clarify the results of the study. I see the methods presented extensively, also the objectives are discussed, but the results should also be presented more explicitly.

corrected comments

  1. Introduction - Has the topic been addressed before? You should bring forth papers which tackle similar topics, so that you outline the significance of the present study, why it is needed. The literature review is very, very scarce.

corrected comments

  1. Methodology - Are there any limitations / weaknesses of the model used?

There are no weaknesses

  1. The conclusions are very vague, they should get more consistency, underlining the added value of the paper.

Worked out and added conclusions

Reviewer 4 Report

The abstract does not discuss the purpose of the research and its results.

In the "Introduction" you need to clearly define the purpose of the research, while in the "Concluding Remarks" section it is necessary to draw specific conclusions in reference to that. The goal can be guessed from the hypothesis, however, the absence of the goal results in too general inference. The results should be discussed more extensively.

The relationship between the content of the paper and its title is not clear, and as a result the study is not consistent.

The paper lacks a chapter embedding the conducted analysis in the economic theory. Perhaps this is due to the fact that the study is not of an economic nature. Demonstrating the relationship with the theory of economics is advisable due to the profile of the journal. If there is no such relationship, perhaps it would be better to publish the paper in a journal with a sociological profile.

Apart from logit models, did the authors consider the usage of other research methods? This is not a complaint, but a suggestion to discuss this issue in more detail.

The sentence in line 158: "When building a model, it is possible that some of these categories of responses will be included in the model, and some will not be included" requires clarification.

I am not convinced that the sentence "This program is one of the most popular in the former Soviet Union" (line 137) is the best justification for using the SPSS program.

Please, remember that the econometric methods used are only a research tool, not the aim of the study. Without embedding the analysis in the theory and discussing the results in relation to the frame of reference (e.g. economic or social one, labor market needs), the paper is more like a research report rather than a scientific text.

Author Response

Reviewers 4

  1. In the "Introduction" you need to clearly define the purpose of the research, while in the "Concluding Remarks" section it is necessary to draw specific conclusions in reference to that. The goal can be guessed from the hypothesis, however, the absence of the goal results in too general inference. The results should be discussed more extensively.

Goal, result added, conclusions expanded.

  1. The relationship between the content of the paper and its title is not clear, and as a result the study is not consistent.

The title of the article indicates that a multidimensional logistic model is necessary to predict the employment of graduates of higher educational institutions of the Republic of Kazakhstan. The article shows what affects the employment of students.

  1. The paper lacks a chapter embedding the conducted analysis in the economic theory. Perhaps this is due to the fact that the study is not of an economic nature. Demonstrating the relationship with the theory of economics is advisable due to the profile of the journal. If there is no such relationship, perhaps it would be better to publish the paper in a journal with a sociological profile.

Have the authors considered using other research methods besides logit models? This is not a complaint, but a proposal to discuss this issue in more detail.

  1. Apart from logit models, did the authors consider the usage of other research methods? This is not a complaint, but a suggestion to discuss this issue in more detail.

Text relating to this remark has been added. A text was added about the broken model.

  1. The sentence in line 158: "When building a model, it is possible that some of these categories of responses will be included in the model, and some will not be included" requires clarification.

I'm not convinced that the sentence "This program is one of the most popular in the former Soviet Union" (line 137) is the best justification for using SPSS.

  1. I am not convinced that the sentence "This program is one of the most popular in the former Soviet Union" (line 137) is the best justification for using the SPSS program.

This text has been removed from the article.

  1. Please, remember that the econometric methods used are only a research tool, not the aim of the study. Without embedding the analysis in the theory and discussing the results in relation to the frame of reference (e.g. economic or social one, labor market needs), the paper is more like a research report rather than a scientific text.

Text relating to this remark has been added.

Round 2

Reviewer 1 Report

Dear Authors,

A significant improvement has been made in the study. However, the connection to the scientific literature is still weak. Without it, the scientific contribution is not justified. I can accept that for policymakers and HEI managers, the results can be interesting in their current form, but not for the audience of a scientific paper.

The more detailed comments:

The introduction section is like a draft listing too many research objectives without justification or scientific grounding.

Details about the pilot study are still missing (when was it conducted, who were the participants).

Page 3, last paragraph: „Most of the available data in articles and publications…”, „In recent years, the results obtained from surveys of…”, „All available information suggests…”: Have the authors conducted systematic literature research justifying this statement? If yes, please provide the details.

The last paragraph of section 2 belongs to the methodology.

Lines 168-172: the authors estimated a 15% ratio of students taking part in international mobility programs for every student in Kazakhstan based on one university. Why do they assume that this university represents the whole country? Are country-level data unavailable (e.g. from the ministry responsible for HEI)? More importantly: why was this estimation necessary if they had received student data from the selected universities as they reported below?

How was the randomness of the selection of universities ensured?

How were the questionnaire items developed? Are they connected to the scientific literature somehow?

The variables are still not defined appropriately because the response options are unreported. For example, "The quality of education” is a dummy variable, but we do not know what 0 and 1 stand for. Bad and Good? Acceptable, Not acceptable? Exceptional and Average?

A reference is missing from line 225.

Lines 232-236: I still do not accept linear correlation as the appropriate test. I know that correlation is calculable on dummy variables. I agree that this is the same as the chi-square calculated for binary variables. However, from an interpretative point of view, it is not appropriate to use it, since linear correlation measures the expected increase in variable b if variable b increases by one unit. In the case of binary variables, there is no “increase”. Chi-square and Cramer V are the test statistics the authors need. The statement “Since the variables are binary, that is, they take on values either 0 or 1, then it is correct to calculate the correlation coefficients for such variables” does not eliminate my concerns. On the other hand, I can accept a regression model with dummy independent variables.

Line 312: “3.Y” és “Table 3. ” and “Y”.

Lines 315-327 have still not contributed to the manuscript, since it is obvious how a regression equation can be used for prediction.

Please, consider using “.” Instead of “,” in decimals, and “×” or “∙” instead of “*” in equations. On my part, I would also use the random error term.

After adding a proper scientific basis, I still think the study would be worth publishing.

Author Response

Responses to the reviewer's note

A significant improvement has been made in the study. However, the connection to the scientific literature is still weak. Without it, the scientific contribution is not justified. I can accept that for policymakers and HEI managers, the results can be interesting in their current form, but not for the audience of a scientific paper.

The more detailed comments:

The introduction section is like a draft listing too many research objectives without justification or scientific grounding.

Details about the pilot study are still missing (when was it conducted, who were the participants).

A pilot study of 20 questionnaires was previously conducted and took place on May 2, 2020 and was held among M. Auezov SKU university students.

Page 3, last paragraph: „Most of the available data in articles and publications…”, „In recent years, the results obtained from surveys of…”, „All available information suggests…”: Have the authors conducted systematic literature research justifying this statement? If yes, please provide the details.

The last paragraph of section 2 belongs to the methodology.

Lines 168-172: the authors estimated a 15% ratio of students taking part in international mobility programs for every student in Kazakhstan based on one university. Why do they assume that this university represents the whole country? Are country-level data unavailable (e.g. from the ministry responsible for HEI)? More importantly: why was this estimation necessary if they had received student data from the selected universities as they reported below?

Among the 14 universities of Kazakhstan that participated in the rating of the international rating agency Quacquarelli Symonds (QS), the Auezov University is in the top three. It is the oldest and largest university in the Turkestan region. Due to the fact that this university has the largest number of students in a wide range of humanities and technical specialties, which positively affects the representativeness of such a university and because of the convenience and available opportunities, the students of this university were identified by the basic sample. The survey control technique was also easier to provide on-site within the walls of the university.

How was the randomness of the selection of universities ensured?

The rest of the universities that were included in the sample were randomly selected from the list of universities in Kazakhstan and a survey was organized at the locations of the selected universities.

How were the questionnaire items developed? Are they connected to the scientific literature somehow?

Here you need to specify which questionnaires were used to compile the questionnaire, which questionnaires of other authors?

The variables are still not defined appropriately because the response options are unreported. For example, "The quality of education” is a dummy variable, but we do not know what 0 and 1 stand for. Bad and Good? Acceptable, Not acceptable? Exceptional and Average?

The complete questionnaire with the categories of answers is presented in Appendix 1, since it will take too much space in the article to fully present the questions with all the categories of answers to them.

A reference is missing from line 225.

Done, a reference is defined.

Lines 232-236: I still do not accept linear correlation as the appropriate test. I know that correlation is calculable on dummy variables. I agree that this is the same as the chi-square calculated for binary variables. However, from an interpretative point of view, it is not appropriate to use it, since linear correlation measures the expected increase in variable b if variable b increases by one unit. In the case of binary variables, there is no “increase”. Chi-square and Cramer V are the test statistics the authors need. The statement “Since the variables are binary, that is, they take on values either 0 or 1, then it is correct to calculate the correlation coefficients for such variables” does not eliminate my concerns. On the other hand, I can accept a regression model with dummy independent variables.

The mention of the Pearson linear correlation has been removed from the article. Chi-square and Cramer V are abandoned.

Line 312: “3.Y” és “Table 3. ” and “Y”.

It is not clear, there is no such thing on this line.

Lines 315-327 have still not contributed to the manuscript, since it is obvious how a regression equation can be used for prediction.

This remark is also unclear. If the reviewer insists, this example with the calculation of the probability of a positive answer to the question about the benefits of participation in academic mobility programs can be removed altogether.

Please, consider using “.” Instead of “,” in decimals, and “×” or “∙” instead of “*” in equations. On my part, I would also use the random error term.

Corrected throughout the text

After adding a proper scientific basis, I still think the study would be worth publishing.

Reviewer 3 Report

I agree to the paper being published.

Author Response

Thank you so much

Reviewer 4 Report

The paper looks better. The authors took into account most of the recommendations.

Author Response

Thank you so much

Round 3

Reviewer 1 Report

The authors just left the following three comments unanswered.

However, the connection to the scientific literature is still weak. Without it, the scientific contribution is not justified. I can accept that for policymakers and HEI managers, the results can be interesting in their current form, but not for the audience of a scientific paper.

The introduction section is like a draft listing too many research objectives without justification or scientific grounding.

Page 3, last paragraph: „Most of the available data in articles and publications…”, „In recent years, the results obtained from surveys of…”, „All available information suggests…”: Have the authors conducted systematic literature research justifying this statement? If yes, please provide the details.

The problems mentioned above remain unresolved. The research is still not anchored in the scientific literature and the vague (“most of the available”, surveys”, “all available”) references to the unspecified general literature mentioned above remain inadequate.

In my previous review, I may not have been clear when I mentioned that the left-hand side of equation (2) is "3.Y". That is incorrect because the "Table 3" at the end of line 322 is "3." has slipped through. It is still not corrected, and the authors responded “It is not clear, there is no such thing on this line.”. I still can see it.

All other responses and changes are acceptable.

Author Response

The authors just left the following three comments unanswered.

Dear reviewers, I'm sorry that I missed these 3 comments, because there were a lot of comments and somehow we missed them in a hurry

“However, the connection to the scientific literature is still weak. Without it, the scientific contribution is not justified. I can accept that for policymakers and HEI managers, the results can be interesting in their current form, but not for the audience of a scientific paper.”

We have tried to update the literature

 “The introduction section is like a draft listing too many research objectives without justification or scientific grounding.”

We have corrected the introduction according to the comment of the reviewer and have written more extensively

“Page 3, last paragraph: „Most of the available data in articles and publications…”, „In recent years, the results obtained from surveys of…”, „All available information suggests…”: Have the authors conducted systematic literature research justifying this statement? If yes, please provide the details.”

The survey was conducted; the survey was conducted by me

The problems mentioned above remain unresolved. The research is still not anchored in the scientific literature and the vague (“most of the available”, surveys”, “all available”) references to the unspecified general literature mentioned above remain inadequate.

"We have expanded the literature review to include more recent studies and to provide a more comprehensive context for our research."

In my previous review, I may not have been clear when I mentioned that the left-hand side of equation (2) is "3.Y". That is incorrect because the "Table 3" at the end of line 322 is "3." has slipped through. It is still not corrected, and the authors responded “It is not clear, there is no such thing on this line.”. I still can see it.

This comment has been corrected. The formulas have been re-numbered.

All other responses and changes are acceptable.

I am very grateful for your patience and understanding, and we apologize for the omission of several comments earlier, we have tried to correct all the comments